# Annular Ligament Instability in Lateral Elbow Pathology: Objective Confirmation Through a Cadaveric Study

**DOI:** 10.3390/muscles4030039

**Published:** 2025-09-15

**Authors:** Daniel Berlanga de Mingo, Guillem Paz Ramírez, Arnau Moreno Garcia, Maria Tibau Alberdi, Diana Noriego Muñoz, Miguel Pérez Abad, Giacomo Rossettini, Jorge Hugo Villafañe, César Abellán Miralles, Montserrat del Valle Jou, Àngel Ferreres Claramunt, Alfonso Rodríguez Baeza

**Affiliations:** 1Department of Shoulder, Knee and Elbow Arthroscopy, Hospital Asepeyo Sant Cugat, 08174 Barcelona, Spain; dberlangademingo@asepeyo.es (D.B.d.M.); cabellanmiralles@asepeyo.es (C.A.M.); 2Department of Anatomy and Morphologic Science, Faculty of Medicine, Universitat Autònoma de Barcelona, 08193 Cerdanyola del Vallès, Spain; mariatibaualberdi@gmail.com (M.T.A.); miguelp.abad@gmail.com (M.P.A.); montsedelvallejou@gmail.com (M.d.V.J.); angelferreres@institut-kaplan.com (À.F.C.); alfonso.rodriguez@uab.cat (A.R.B.); 3Department of Hand and Elbow, Hospital de Mataró, 08301 Barcelona, Spain; pazguillem@gmail.com; 4Department of Hand and Elbow, Hospital Josep Trueta, 17007 Girona, Spain; amorenogarcia93@gmail.com (A.M.G.);; 5Department of Hand and Elbow, Institut Kaplan, 08034 Barcelona, Spain; 6Department of Physiotherapy, Faculty of Medicine, Health and Sports, Universidad Europea de Madrid, 28670 Villaviciosa de Odón, Spain; 7Department of Hand and Elbow, Hospital de l’Esperit Sant, 08923 Santa Coloma de Gramenet, Spain

**Keywords:** anular ligament, elbow instability, cadaveric study

## Abstract

Background: The annular ligament is a key secondary stabilizer of the elbow, but its biomechanical behavior during forearm rotation has not been objectively quantified. This study aimed to assess interindividual variability in annular ligament tension, validate prior arthroscopic observations, and explore associations with chondral lesions in the lateral elbow compartment. Methods: In this cross-sectional anatomical study, 25 cadaveric upper limbs were analyzed following standardized dissection, preserving ligamentous and muscular integrity. Ligament displacement was measured using a custom mechanical apparatus and high-precision digital micrometer in neutral, 60° pronation, and 60° supination positions under axial tractions of 1, 2, and 3 kg. Ulnar length and presence of chondral lesions were also recorded. Results: Maximal ligament displacement occurred in supination in 80% of specimens (mean: 1.23 mm at 3 kg; range: 0.30–2.87 mm), indicating considerable interindividual variation. Significant displacement differences were observed between all forearm positions across load levels (*p* < 0.001). Chondral lesions were identified in three specimens with marked ligament laxity and reduced radial head coverage. Conclusions: This study provides the first objective evidence of annular ligament tension variability during forearm rotation. Ligament laxity may contribute to lateral elbow instability and cartilage degeneration, supporting the ligament’s role as a secondary stabilizer.

## 1. Introduction

Elbow stability is maintained through the coordinated integrity of osseous structures, ligamentous elements, and active muscular control. Primary static stabilizers include the humeroulnar joint, the medial collateral ligament (MCL), and the lateral collateral ligament complex (LCL), while secondary stabilizers comprise the humeroradial joint, the joint capsule, and the surrounding musculature, particularly the flexor-pronator and extensor-supinator groups of the forearm [1,2]. These muscle groups not only contribute to movement generation but also serve as dynamic stabilizers, providing joint compression and compensatory restraint under valgus or varus stress, especially during functional tasks and load-bearing positions [2,3,4]. Their anatomical proximity and fascial continuity with ligament insertions further support their biomechanical role.

Within the lateral compartment of the elbow, the LCL consists of four distinct yet functionally integrated components: the annular ligament, the radial collateral ligament (RCL), the lateral ulnar collateral ligament (LUCL), and the accessory lateral collateral ligament (ALCL) [5]. These elements, in synergy with the extensor-supinator musculature, maintain static and dynamic lateral elbow stability. Disruption of any component may compromise joint congruity and result in symptomatic instability.

The RCL and LUCL originate from a common isometric point on the inferior aspect of the lateral epicondyle, enabling consistent tension across elbow flexion and extension. The LUCL inserts on the supinator crest of the proximal ulna, functioning as a principal restraint to varus stress and posterolateral rotatory instability. The annular ligament, which encircles the radial head and attaches to the anterior and posterior margins of the radial notch of the ulna, works in concert with the RCL to secure radial head articulation. The ALCL, though variably present, originates from the annular ligament and blends with the LUCL at the supinator crest [6,7].

Forearm pronation and supination require precise articulation among the radial head, the radial notch, and the capitellum. During supination, the radial head engages posteriorly with the capitellum, whereas pronation induces anterior translation—movements constrained by the annular ligament [8,9]. Though traditionally regarded as a static stabilizer confining the radial head, the annular ligament has recently been implicated in broader biomechanical roles within lateral elbow stability [4]. Earlier assumptions of uniform anterior-posterior fiber tensioning during rotation [10,11] have been re-evaluated in light of recent arthroscopic studies revealing notable interindividual variability.

Clinical and biomechanical evidence suggests that elongation or incompetence of the annular ligament may result in proximal radioulnar joint instability, a condition often underdiagnosed and misattributed to chronic lateral epicondylalgia. In patients with symptomatic minor instability of the lateral elbow (SMILE), arthroscopic evaluation has revealed morphologic variations in radial head containment and ligament tension, correlating with radiocapitellar incongruity and chondral wear [12]. Furthermore, progressive elongation of the annular ligament and RLCL has been associated with pathological anteroposterior shifting of the radial head, reducing contact with the inner ligament surface and further compromising joint congruency and function [13].

Despite increasing clinical recognition, no cadaveric study to date has quantitatively evaluated the tension behavior or anatomical variability of the annular ligament during controlled forearm rotation [14]. Such investigations are crucial to validate arthroscopic findings and to enhance understanding of ligamentous behavior under functional loading.

The aim of this study is to objectively assess the biomechanical behavior and anatomical variability of the annular ligament in a cadaveric model. Specifically, we sought to corroborate previous arthroscopic observations, investigate the relationship between ligament displacement and chondral lesions, and clarify the functional contribution of the annular ligament during forearm rotation. By addressing these questions, this study contributes to the expanding knowledge of lateral elbow pathology and highlights the relevance of musculoligamentous interactions in maintaining elbow stability.

## 2. Materials and Methods

### 2.1. Study Design and Ethical Approval

A cross-sectional anatomical study was conducted using 25 human cadaveric upper limbs obtained from adult donors through the university’s morgue, in accordance with protocols approved by the Ethics Committee for Animal and Human Research of the Universitat Autònoma de Barcelona (Procedure No. 2904, approved on 27 March 2015). This protocol authorizes the use of donated bodies and anatomical specimens for teaching, postgraduate training, research, and continuing education purposes. All donors had previously consented to the donation of their bodies for scientific and educational purposes, either personally or through next-of-kin consent as required by applicable regulations [15]. Donor data are stored in a secure institutional database and handled in compliance with current data protection laws. Prior to use, all specimens were anonymized and serologically screened for hepatitis B, hepatitis C, and HIV. Only information regarding the donor’s age and sex was accessible to the research team [16].

The study was approved by the Clinical Research Ethics Committee of Catalonia (CEIC), which issued a favorable opinion for its execution. All procedures were conducted in compliance with the ethical guidelines of the Universitat Autònoma de Barcelona and national regulations on body donation.

### 2.2. Specimens and Dissection

The anatomical specimens consisted of upper limbs preserved from the mid-humerus to the hand, stored at −40 °C, and thawed at room temperature for 24 h prior to dissection. The dissection involved a longitudinal lateral approach to the distal humerus, carefully avoiding the lateral epicondyle to preserve tendon and ligament attachments. Blunt dissection was performed anteriorly and medially to expose the joint capsule while maintaining the integrity of the overlying musculature, including partial release of the brachioradialis muscle. A limited anterior capsulotomy was performed over the humeral capitellum to expose the cranial portion of the annular ligament at 90° of elbow flexion. Ulnar length was measured to allow consistent positioning across specimens.

This preservation protocol has been validated in previous cadaveric studies as maintaining ligamentous tissue integrity without significantly altering biomechanical properties. To further minimize the risk of tissue alteration, specimens were thawed only once at room temperature, and no specimen was subjected to repeated freeze–thaw cycles.

### 2.3. Measurement Apparatus and Protocol

Each specimen was mounted on a custom-designed wooden apparatus that secured the humerus and ulna at 90° of elbow flexion while permitting free pronation-supination of the forearm. An electronic micrometer (Wisamic^®^, Model C-10, ASIN B074TD82N8; Wisamic, Shenzhen, China), with 1 µm precision and a 12.7 mm maximum range, was positioned anterior to the annular ligament. The device’s hooked probe was inserted from proximal to distal between the annular ligament and the radial head, contacting the apex of the latter.

Fixation to the apparatus was achieved using the following protocol (Figure 1):

Two 3.2 mm Kirschner wires were transversely inserted through the ulna: one at the level of the radial neck (below the supinator crest to avoid damaging the annular ligament and the LCL), and another at the mid-diaphysis, Figure 1a.

A 4.5 mm Kirschner wire was inserted through the humerus and anchored to the support frame, Figure 1b.

A 2 mm Kirschner wire was inserted through a single cortex of the radial diaphysis and connected to a goniometer to allow controlled movement though 60° pronation and 60° supination, Figure 1c.

Ligament displacement was measured in three standardized forearm positions (neutral, 60° pronation, and 60° supination), with the order of measurement randomized to minimize systematic bias. Axial traction forces of 1, 2, and 3 kg were applied using a calibrated spring dynamometer (Figure 2). Heavier loads were avoided due to preliminary evidence of potential ligament injury. The choice of traction levels was informed by pilot testing conducted to determine the maximum safe load; in these trials, failure occurred in one specimen at 4 kg, leading us to restrict the experimental range to 3 kg or less. To our knowledge, no previous studies have directly quantified the physiological loading borne specifically by the annular ligament during daily activities. Accordingly, the selected loads represent a methodological compromise between safety and the need to approximate a physiologically relevant spectrum of tensile forces.

### 2.4. Validation and Data Collection

The measurement system was validated by repeated intra-observer assessments on the first three specimens, showing high consistency. Calibration confirmed that the mechanical resistance of the setup introduced a negligible error (approximately 350 g required to achieve full micrometer displacement of 12.7 mm), which was considered acceptable in light of the applied loads and the small magnitude of measured displacements.

Annular ligament laxity was also qualitatively assessed during dissection by applying gentle caudal traction with forceps. In the three specimens that exhibited the greatest displacement values on micrometer testing, manual evaluation similarly revealed marked ligamentous laxity. This concordance supports the validity of the quantitative measurements and reinforces the reliability of the observed inter-specimen differences. Chondral lesions in the capitellum and radial head were documented when present. All data were recorded in a Microsoft Excel^®^ database for subsequent statistical analysis.

### 2.5. Statistical Analysis

Displacement data were summarized as means, standard deviations, and ranges. The Shapiro–Wilk test was used to verify the normality of distributions. As values followed a normal distribution, parametric analyses were applied. Differences in ligament displacement across forearm positions (neutral, pronation, supination) under each load condition (1, 2, 3 kg) were assessed using repeated-measures analysis of variance (ANOVA). When significant main effects were detected, pairwise comparisons were performed with Tukey adjustment. Statistical significance was set at *p* < 0.05. All analyses were conducted using IBM SPSS Statistics, version 28 (IBM Corp., Armonk, NY, USA).

## 3. Results

Twenty-five cadaveric upper limb specimens (12 male, 13 females; 16 right, 9 left) were examined, with a mean age at death of 87.4 years (range: 70–100). Ulnar length was consistently 26 cm, except in three male specimens measuring 28 or 29 cm.

Annular ligament displacement under 3 kg of axial traction in the neutral position revealed considerable interindividual variability (Table 1). Most elbows (*n* = 9) exhibited displacements between 1.0 and 1.5 mm, while only three specimens surpassed the 2.0 mm threshold, suggesting distinct biomechanical phenotypes with varying ligamentous compliance.

When traction loads were progressively increased from 1 to 3 kg, ligament separation increased proportionally across all forearm positions (Table 2). Supination consistently produced the greatest displacement, followed by neutral and pronation, indicating a position- and load-dependent response pattern.

Notably, three specimens with ligament displacements >2.0 mm exhibited visible chondral lesions in the radial head and humeral capitellum, as documented in Figure 3. These cases also presented diminished radial head coverage by the annular ligament, suggesting a biomechanical association between excessive ligamentous laxity and articular degeneration.

Statistical analysis confirmed normal distribution of displacement values, allowing for parametric testing. Repeated-measures ANOVA demonstrated significant differences in displacement between pronation, neutral, and supination across all load conditions. Pairwise comparisons using the Tukey correction revealed that supination consistently induced significantly greater displacement (Table 3).

## 4. Discussion

This cadaveric study provides the first objective and quantitative evidence of interindividual variability in annular ligament behavior during forearm pronation and supination. These findings corroborate prior arthroscopic observations and support the hypothesis that differences in ligament tension contribute to the pathogenesis of lateral elbow pain and degenerative joint changes.

Our results show strong concordance with the classification proposed by Abellán-Miralles et al. (2024) [12], who identified three annular ligament types based on intraoperative assessments of tension and radial head coverage. The consistency between our biomechanical data and their clinical observations reinforces the diagnostic relevance of annular ligament evaluation in patients with persistent lateral elbow pain not attributable to classic epicondylalgia.

From a biomechanical perspective, the variability in ligament displacement among specimens reflects differences in viscoelastic tissue properties. This heterogeneity may explain the diverse clinical responses to conservative treatments such as physiotherapy, bracing, or corticosteroid injections. Specifically, individuals with high-tension (type 1) ligaments may benefit from nonoperative stabilization strategies, whereas those with low-tension (type 3) ligaments might require surgical intervention to restore radial head containment and joint congruity [17].

A key finding was the association between displacements exceeding 2 mm and chondral lesions at the radiocapitellar interface, suggesting a potential causal link between annular ligament insufficiency and articular cartilage degeneration [18]. These results align with biomechanical models, indicating that altered joint kinematics lead to focal overload and wear. Given the annular ligament’s role in radial head stabilization during forearm rotation [3], increased laxity likely contributes to microinstability and abnormal load distribution, predisposing the joint to degenerative changes. Although this threshold was not defined a priori, the repeated observation of chondral degeneration in specimens with displacements above 2.0 mm indicates potential clinical significance. We regard this as an exploratory finding that highlights the need for validation in vivo, ideally through advanced imaging techniques, before it can be considered a diagnostic benchmark. Finally, while our data support a causal direction from ligament laxity to chondral injury, the alternative possibility—that pre-existing cartilage damage reduces congruency and secondarily alters ligament tension—cannot be entirely excluded.

The observed proportional increase in displacement during axial traction and supination further highlights the annular ligament’s role as a dynamic stabilizer. Supination appears particularly susceptible to pathological translation, possibly due to reduced posterior fiber tension. Clinically, this may underlie the pain experienced during supinated tasks, such as lifting with the palm facing upward, in patients with suspected ligament laxity.

Our data also contribute to the broader understanding of secondary stabilizers of the elbow. While the LCL is well-established as the primary static stabilizer, our findings underscore the substantial yet often underrecognized contribution of the annular ligament to lateral elbow stability. This is consistent with the work of Traverso et al. (2025) [19], who proposed arthroscopic plication of the annular ligament as a safe and reproducible technique for restoring joint stability in patients with symptomatic minor lateral elbow instability (SMILE). Their approach, targeting the “loose collar sign,” yielded favorable clinical outcomes and highlights the importance of directly addressing ligamentous laxity.

Furthermore, our results are in agreement with those of Vannitamby et al. (2025) [20], who demonstrated in a Monteggia fracture model that annular ligament repair most effectively restored radial head stability, outperforming several reconstructive techniques. Although their focus was on anterior instability, their findings support a broader principle: whenever feasible, preserving or repairing the native annular ligament provides superior biomechanical outcomes compared to reconstruction with autografts or synthetic substitutes. This underscores the clinical imperative of recognizing and preserving ligament integrity during surgical decision-making.

### 4.1. Applications for Clinical Practice and Future Research Lines

These findings underscore the potential of annular ligament tension as a clinically relevant biomechanical parameter. Incorporating advanced imaging modalities—such as high-resolution ultrasound or load-sensitive MRI—could facilitate in vivo quantification of ligament displacement and radial head coverage, improving diagnostic accuracy in cases of suspected microinstability.

Therapeutically, the stratification of patients based on ligament phenotype could guide personalized rehabilitation. High-tension ligaments may benefit from targeted strengthening and load management, while lax ligaments may require surgical interventions such as annular ligament plication or reconstruction—approaches already validated for LUCL pathology and increasingly extended to annular ligament laxity [19].

Future research should aim to define normative displacement ranges across age, sex, and activity levels, particularly in high-demand populations such as athletes or manual laborers. Additionally, prospective clinical studies correlating in vivo biomechanical assessments with patient-reported outcomes and surgical findings are needed to validate the diagnostic and prognostic utility of ligament tension analysis. Computational modeling, including finite element analysis, may further elucidate the mechanical consequences of ligament laxity on elbow joint biomechanics.

### 4.2. Limitations of the Study

Several limitations warrant consideration. First, although the sample size (*n* = 25) is adequate for cadaveric biomechanical research, it is insufficient to perform subgroup analyses based on sex, limb dominance, or ulnar morphology. Larger, demographically diverse samples are needed to explore potential anatomical and physiological modifiers of annular ligament behavior.

Second, adjacent stabilizing structures—particularly the quadrate ligament and the interosseous membrane (notably its central band)—were not preserved. These elements substantially contribute to proximal radioulnar joint stability by distributing loads and constraining excessive translation [17,21]. Their absence in our dissections may have resulted in greater annular ligament displacement than would occur in vivo. Thus, while the isolated preparation allowed for standardized measurements of annular ligament behavior, the results should be interpreted as reflecting the ligament’s performance under reduced stabilizing conditions rather than as exact replicas of intact joint mechanics.

Third, the advanced age of the specimens (mean donor age 87.4 years) may have influenced the mechanical properties observed. Age-related changes could theoretically predispose ligaments either to increased laxity or to stiffening through collagen cross-linking and reduced elasticity. Nonetheless, our results did not deviate from previously published data in younger populations. Indeed, the outcomes were consistent with those reported in our earlier work [12], suggesting that despite potential age-related alterations, the fundamental biomechanical behavior of the annular ligament remains preserved. Even so, the generalizability of these findings to younger or more physically active individuals is limited, underscoring the need for future studies employing younger specimens or in vivo imaging methods.

Finally, technical constraints precluded the simultaneous evaluation of other stabilizers, such as the LCL, which also plays a critical role in lateral elbow stability. Future investigations using more complete anatomical preparations, younger specimens, and in vivo imaging approaches could provide complementary insights and strengthen the external validity of the present findings.

## 5. Conclusions

This study provides the first objective evidence of interindividual variability in annular ligament tension during forearm rotation, confirming and extending previous arthroscopic observations through precise biomechanical measurements. The data reveal distinct ligamentous behavior patterns—ranging from high-tension to redundant configurations—that likely correspond to different clinical phenotypes of lateral elbow pain and instability.

Our findings support a biomechanical link between annular ligament laxity and the presence of chondral degeneration in the radiocapitellar joint, reinforcing the role of the annular ligament as a critical secondary stabilizer of the lateral elbow compartment. This relationship suggests that ligament insufficiency may be an underrecognized contributor to joint microinstability and cartilage overload, particularly in supinated positions.

Furthermore, the observation that the anterior portion of the annular ligament experiences greater tension in pronation than in supination is directly supported by our measurements. Because the humeral condyle was immobilized during testing, the radial head remained constant in the axial plane, and the observed displacement was attributable to movement of the ligament around the radial head under traction. This differential behavior may carry diagnostic value and inform both surgical decision-making and rehabilitation strategies.

Taken together, these results highlight the importance of considering annular ligament integrity in the assessment of lateral elbow stability. Future studies are warranted to validate these findings in vivo, define their clinical applicability, and determine whether ligament displacement assessment can serve as a reliable diagnostic parameter.

## Figures and Tables

**Figure 1 muscles-04-00039-f001:**
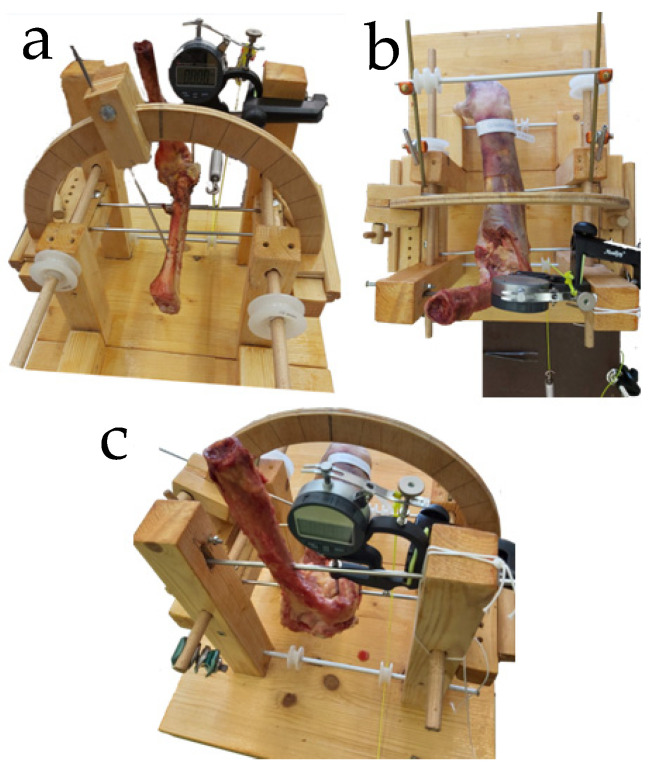
Custom wooden apparatus for specimen fixation and testing. Left elbow setup (**a**); right elbow in pronation (**b**,**c**). (**a**) Ulna: two 3.2 mm K-wires (radial neck, mid-diaphysis). (**b**) Humerus: 4.5 mm K-wire anchored to frame. (**c**) Radius: 2 mm K-wire connected to goniometer (±60° pronation–supination).

**Figure 2 muscles-04-00039-f002:**
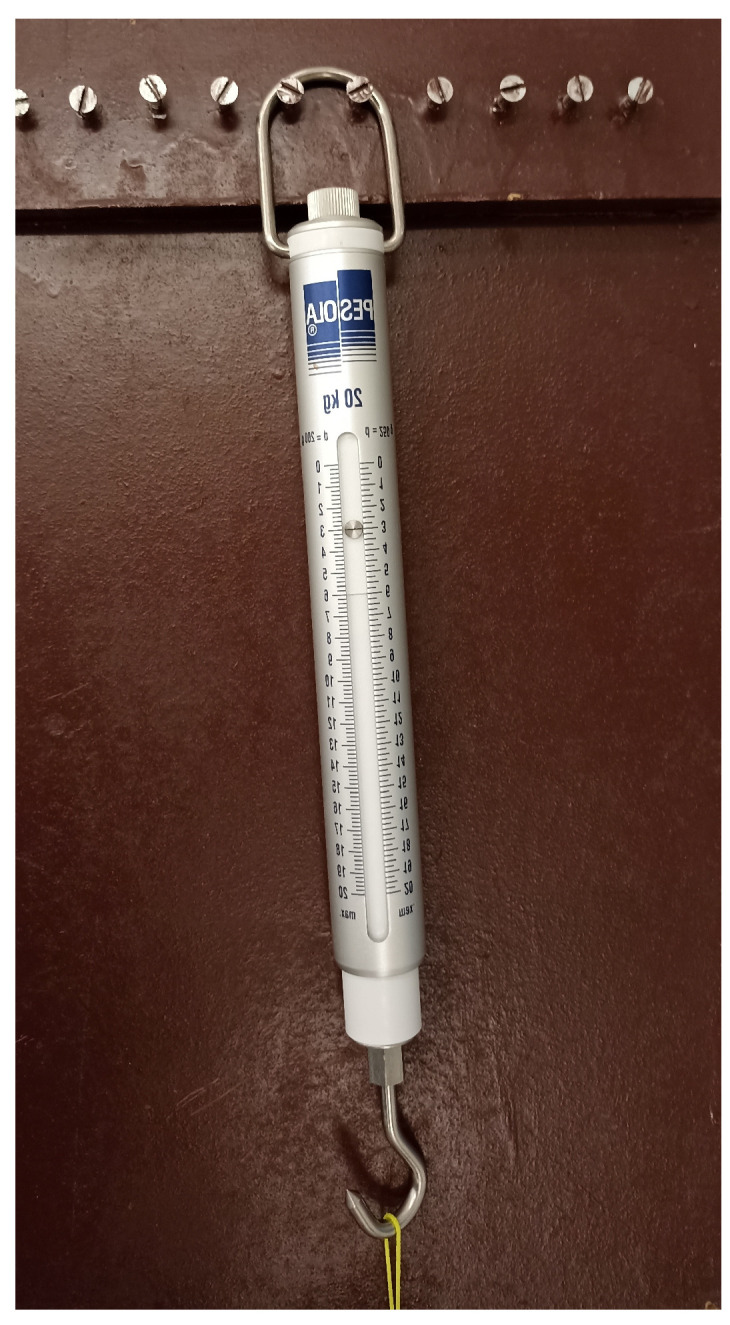
A device used to measure the load applied to the ligament at 2 and 3 kg, respectively.

**Figure 3 muscles-04-00039-f003:**
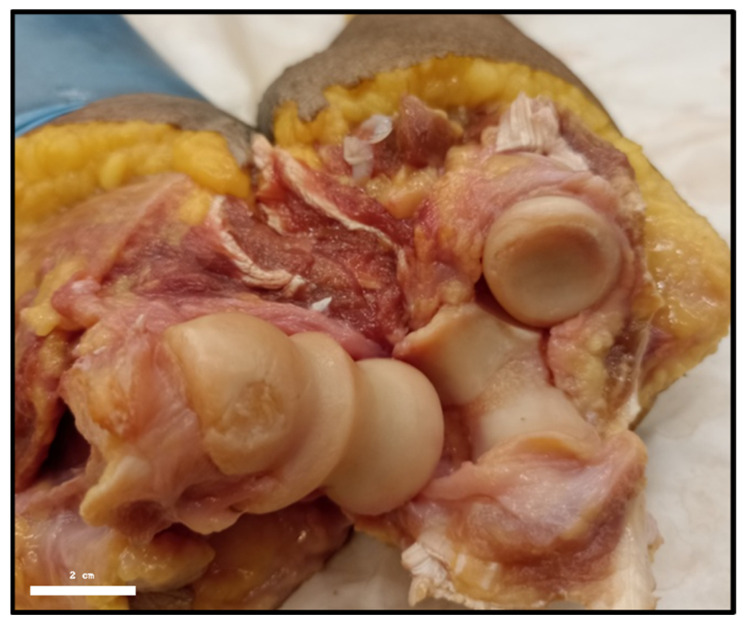
Chondral lesion in the humeral condyle and radial head (Case 25: right elbow, 84-year-old female).

**Table 1 muscles-04-00039-t001:** Annular ligament displacement under 3 kg traction in neutral forearm position (mm).

Displacement Range (mm)	Number of Elbows
0.0–0.5	4
0.5–1.0	5
1.0–1.5	9
1.5–2.0	4
2.0–2.5	1
2.5–3.0	2

**Table 2 muscles-04-00039-t002:** Mean annular ligament displacements (mm) across forearm positions (pronation, neutral, supination) under 1, 2, and 3 kg of axial traction.

Load	Pronation	Neutral	Supination
1 kg	0.1	0.2	0.2
2 kg	0.5	0.5	0.7
3 kg	1.1	1.2	1.4

**Table 3 muscles-04-00039-t003:** Paired comparisons of annular ligament displacement between forearm positions (mm).

Pair	Mean Difference	95% CI (Lower–Upper)	t	df	*p* (2-Tailed)
Supination–Pronation (S1–P1)	0.1 ±0.2	0.0–0.2	2.6	24	0.016
Supination–Neutral (S2–P2)	0.3 ±0.4	0.1–0.4	3.3	24	0.003
Supination–Pronation (S3–P3)	0.4 ±0.5	0.2–0.6	4.2	24	<0.001

Legend: Values represent the mean differences in displacement (mm) between paired forearm positions under progressive traction loads. Supination consistently resulted in significantly greater ligament separation compared to pronation and neutral positions (repeated-measures ANOVA, Tukey-adjusted pairwise comparisons).

## Data Availability

Data (raw) are available upon formal request fromthe corresponding author.

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
