# Peer review of "Annular Ligament Instability in Lateral Elbow Pathology: Objective Confirmation Through a Cadaveric Study"

_muscles, 2025, doi:10.3390/muscles4030039_

Round 1
Reviewer 1 Report
Comments and Suggestions for Authors
Dear Authors,
I sincerely appreciate the effort and scientific quality demonstrated in this manuscript. The study rigorously addresses the biomechanical variability of the annular ligament of the elbow and its implications for lateral instability, providing objective evidence using a cadaveric model. However, the manuscript needs some improvements to strengthen its validity and clinical applicability. Here are my suggestions:
- Figure 1 lacks a scale bar. Please add.
- I strongly recommend dividing the materials and methods section into sections (e.g., study design, specimens, dissection, measuring device, protocol, etc)
- Besides, a statistical analysis section should be added to the materials and methods section.
- It should be described how the lack of other stabilizers (interosseous membrane, quadratic ligament) can influence the measured results.
- In my opinion, I notice an excess of speculative or promotional language in some sentences. For example: Line 297 "...open new avenues for personalized treatment approaches...". It should be softened or supported with more direct evidence.
Best regards.
Author Response
Reviewer’s comments
#1 reviewer :
Comments and Suggestions for Authors
Dear Authors,
I sincerely appreciate the effort and scientific quality demonstrated in this manuscript. The study rigorously addresses the biomechanical variability of the annular ligament of the elbow and its implications for lateral instability, providing objective evidence using a cadaveric model. However, the manuscript needs some improvements to strengthen its validity and clinical applicability. Here are my suggestions:
Figure 1 lacks a scale bar. Please add.
Response: done
I strongly recommend dividing the materials and methods section into sections (e.g., study design, specimens, dissection, measuring device, protocol, etc)
Response: done.
Besides, a statistical analysis section should be added to the materials and methods section.
Response: Done.
It should be described how the lack of other stabilizers (interosseous membrane, quadratic ligament) can influence the measured results.
Response: This point has been addressed in the Discussion – Limitations of the Study section.
In my opinion, I notice an excess of speculative or promotional language in some sentences. For example: Line 297 "...open new avenues for personalized treatment approaches...". It should be softened or supported with more direct evidence.
Response: We have completed. We’ve completed.
Reviewer 2 Report
Comments and Suggestions for Authors
ABSTRACT
I would suggest adding one more keyword related to the use of cadaveric specimens and to the study design, which is cross-sectional.
INTRODUCTION
I recommend adding references to paragraph 2.
DISCUSSION
I recommend adding references to paragraph 4 concerning the functionality of the annular ligament. In this same paragraph, you mention an association between displacement greater than 2 mm and chondral lesions. What do you believe is the direction of causality: does increased laxity raise the likelihood of chondral injury, or is it the other way around?
Line 167: replace “lateral colateral ligament complex” with its acronym LCL, as indicated in the Introduction.
MATERIAL AND METHODS
If the assessment of annular ligament laxity was replicated from another study, I suggest adding the corresponding reference in line 277.
Author Response
#2 reviewer :
Comments and Suggestions for Authors
ABSTRACT
I would suggest adding one more keyword related to the use of cadaveric specimens and to the study design, which is cross-sectional.
Response: done.
INTRODUCTION
I recommend adding references to paragraph 2.
Response: done.
DISCUSSION
I recommend adding references to paragraph 4 concerning the functionality of the annular ligament. In this same paragraph, you mention an association between displacement greater than 2 mm and chondral lesions. What do you believe is the direction of causality: does increased laxity raise the likelihood of chondral injury, or is it the other way around?
Line 167: replace “lateral colateral ligament complex” with its acronym LCL, as indicated in the Introduction.
Response: done, thanks.
MATERIAL AND METHODS
If the assessment of annular ligament laxity was replicated from another study, I suggest adding the corresponding reference in line 277.
Response: done
Reviewer 3 Report
Comments and Suggestions for Authors
This is a high-quality study that provides the first objective, quantitative data on the behavior of the elbow's annular ligament during forearm rotation. Using a sound and precise methodology, the research established that ligament looseness varies significantly among individuals, is greatest when the forearm is supinated (palm up), and that excessive laxity (over 2.0 mm) is strongly associated with cartilage damage. These well-supported conclusions have substantial clinical implications, offering a new path for diagnosing lateral elbow pain and developing personalized treatments based on a patient's specific ligament characteristics. To further clarify the study's context and findings, I would appreciate your comments on the following points:
- The mean donor age was 87.4 years. Could you elaborate on how you believe age-related tissue degeneration might have influenced the results? For instance, would you expect older ligaments to be more lax due to degeneration, or stiffer due to other age-related changes?
- The methods state that specimens were stored at -40°C and thawed prior to use. Could you comment on whether this deep-freezing and subsequent thawing cycle could have influenced the biomechanical properties (e.g., elasticity, tensile strength) of the ligaments, and therefore potentially impacted the displacement results?
- The study notes an association between >2.0 mm of displacement and chondral lesions. Was this an observation made post-hoc, or was there an a priori hypothesis for this cutoff? Do you believe this value holds potential as a future diagnostic benchmark?
- Could you explain the rationale for selecting the specific axial traction loads of 1, 2, and 3 kg? How do you estimate these forces relate to the physiological loads experienced by the elbow during daily activities?
- You mention qualitatively assessing laxity via "manual caudal traction with forceps". How did these qualitative findings correlate with the quantitative measurements from the micrometer?
- In the conclusions, you state that the "anterior portion of the annular ligament experiences greater tension in pronation". As the measurement was of total radial head displacement, this appears to be an inference. Could you clarify the basis for this conclusion?
Author Response
#3 reviewer :
Comments and Suggestions for Authors
This is a high-quality study that provides the first objective, quantitative data on the behavior of the elbow's annular ligament during forearm rotation. Using a sound and precise methodology, the research established that ligament looseness varies significantly among individuals, is greatest when the forearm is supinated (palm up), and that excessive laxity (over 2.0 mm) is strongly associated with cartilage damage. These well-supported conclusions have substantial clinical implications, offering a new path for diagnosing lateral elbow pain and developing personalized treatments based on a patient's specific ligament characteristics. To further clarify the study's context and findings, I would appreciate your comments on the following points:
- The mean donor age was 87.4 years. Could you elaborate on how you believe age-related tissue degeneration might have influenced the results? For instance, would you expect older ligaments to be more lax due to degeneration, or stiffer due to other age-related changes?
Response: Although it could be expected that advanced age would result in greater ligamentous laxity due to degenerative changes, our findings did not show a clear deviation from previous results. In fact, the outcomes were consistent with those reported in our earlier work (reference 12), as emphasized in the second paragraph of the Discussion. This suggests that, despite potential age-related alterations, the fundamental biomechanical behavior of the annular ligament appears to be preserved.
- The methods state that specimens were stored at -40°C and thawed prior to use. Could you comment on whether this deep-freezing and subsequent thawing cycle could have influenced the biomechanical properties (e.g., elasticity, tensile strength) of the ligaments, and therefore potentially impacted the displacement results?
Response: Previous cadaveric studies have demonstrated that deep-freezing at −40°C followed by controlled thawing allows adequate preservation of ligamentous tissue without significantly altering its mechanical properties. Our results align with those reported in studies on living patients, as mentioned in our response to Question 1, which further reinforces this interpretation. In addition, we carefully standardized the thawing protocol, allowing specimens to thaw at room temperature and ensuring that each specimen was only subjected to a single freeze–thaw cycle, thereby minimizing the risk of tissue alteration.
- The study notes an association between >2.0 mm of displacement and chondral lesions. Was this an observation made post-hoc, or was there an a priori hypothesis for this cutoff? Do you believe this value holds potential as a future diagnostic benchmark?
Response: Chondral lesions were systematically examined after the dissection of each specimen. The association with displacement values greater than 2.0 mm emerged during this process. While this threshold was not defined a priori, the consistency of this finding suggests that it may hold clinical relevance. We consider this observation promising but recognize that further refinement and validation in studies involving living patients will be necessary before it can be proposed as a diagnostic benchmark.
- Could you explain the rationale for selecting the specific axial traction loads of 1, 2, and 3 kg? How do you estimate these forces relate to the physiological loads experienced by the elbow during daily activities?
Response: Prior to initiating the formal study, we conducted pilot testing to determine the maximal load that could be safely applied without causing structural damage. In one specimen, failure occurred at 4 kg, which led us to limit the applied loads to 1, 2, and 3 kg in order to avoid excessive stress. To the best of our knowledge, there are no published studies that directly quantify the physiological loading borne specifically by the annular ligament during daily activities. Therefore, our selection of loads was based on a balance between safety and the need to explore a physiologically relevant range.
- You mention qualitatively assessing laxity via "manual caudal traction with forceps". How did these qualitative findings correlate with the quantitative measurements from the micrometer?
Response: In the three specimens that demonstrated the greatest displacement values with the micrometer, qualitative manual testing also revealed evident ligamentous laxity. This concordance supports the validity of our quantitative measurements and reinforces the reliability of the observed differences between specimens.
- In the conclusions, you state that the "anterior portion of the annular ligament experiences greater tension in pronation". As the measurement was of total radial head displacement, this appears to be an inference. Could you clarify the basis for this conclusion?
Response: All measurements were performed with the humeral condyle held immobile, thereby maintaining the radial head in a constant position in the axial plane. The observed displacement was therefore attributable to the movement of the annular ligament around the radial head under traction. For this reason, we consider our conclusion—that the anterior portion of the annular ligament is subjected to greater tension in pronation—directly supported by the data rather than inferred.
Round 2
Reviewer 1 Report
Comments and Suggestions for Authors
Dear Authors,
i would like to thank you for the thorough revision of the manuscript. All the suggested improvements have been satisfactorily addressed. I consider the manuscript substantially improved.
Best.
Reviewer 2 Report
Comments and Suggestions for Authors
No further comments
Reviewer 3 Report
Comments and Suggestions for Authors
The authors have substantively addressed the main methodological and interpretive concerns you raised: they clarified the freeze-thaw protocol, justified the traction-load choice, explicitly stated that the 2.0 mm threshold was an exploratory/post-hoc observation, confirmed manual/quantitative concordance, clarified the immobilization that supports the pronation-tension conclusion, and expanded the Limitations section.
As a minor suggestion, please increase the font size of the text above scale-bar label in Figure 2 to improve readability.